# Increased scalability and sequencing quality of an epigenetic age prediction assay

**Benjamin Mayne**[1]*, **David Chandler**[2], **Christopher Noune**[3], **Thomas Espinoza**[4], **David Roberts**[5], **Chloe Anderson**[1], **Oliver Berry**[1]

1 Environomics Future Science Platform, Commonwealth Scientific and Industrial Research Organisation (CSIRO), Indian Ocean Marine Research Centre, Crawley, Western Australia, Australia, 2 Australian Genome Research Facility, Perth, WA, Australia, 3 Australian Genome Research Facility, Melbourne, VIC, Australia, 4 Burnett Mary Regional Group (BMRG), Bargara, QLD, Australia, 5 Seqwater, Ipswich, QLD, Australia

* benjamin.mayne@csiro.au

**Data Availability Statement:** Fastq files are publicly available at https://data.csiro.au/collection/csiro:58058.

**Funding:** This study was funded by the CSIRO Environomics Future Science Platform. The

## Abstract

Epigenetic ageing in a human context, has been used to better understand the relationship between age and factors such as lifestyle and genetics. In an ecological setting, it has been used to predict the age of individual animals for wildlife management. Despite the importance of epigenetic ageing in a range of research fields, the assays to measure epigenetic ageing are either expensive on a large scale or complex. In this study, we aimed to improve the efficiency and sequencing quality of an existing epigenetic ageing assay for the Australian Lungfish (*Neoceratodus forsteri*). We used an enzyme-based alternative to bisulfite conversion to reduce DNA fragmentation and evaluated its performance relative to bisulfite conversion. We found the sequencing quality to be 12% higher with the enzymatic alternative compared to bisulfite treatment (p-value < 0.01). This new enzymatic based approach, although currently double the cost of bisulfite treatment can increases the throughput and sequencing quality. We envisage this assay setup being adopted increasingly as the scope and scale of epigenetic ageing research continues to grow.

## Introduction

Epigenetic ageing and clocks have been studied and used in a wide range of biomedical and ecological research applications [1,2]. This field of research has assisted in the study of age related diseases and the role of environmental factors [3,4], and wildlife population management [5,6]. In human and most mammalian epigenetic ageing research, an Illumina methylation array or a custom array (HorvathMammalMethylChip40) is typically used [7–12]. This custom methylation array is highly effective in mammalian ageing research as it has probes for highly conserved and age associated cytosine-phosphate-guanine (CpG) sites. However, since it is mammalian specific, it cannot be used for non-mammalian species. Furthermore, in classes of vertebrates, such as bony fish, the most speciose group of vertebrates [13], the prospect of developing a universal fish array is low, due to a large evolutionary divergence between species [14,15]. Subsets of species that are similar on the genomic level could have arrays developed, however evolutionary divergent species may not be captured. Although other

funders had no role in study design, data collection and analysis, decision to publish, or preparation of the manuscript.

**Competing interests:** The authors have declared that no competing interests exist.

methylation arrays can be made similar to the one commonly used in humans [7]. This can be expensive and is unlikely to occur for other species. Therefore, it is important to be able to develop epigenetic ageing assays that can be applied to any species of interest.

In many wild species, alternatives to array-based technology have been used to develop epigenetic age assays [5,6,16–19]. One of these methods is a multiplex PCR followed by second-generation sequencing. The two important parts of this process from tissue to epigenetic age is the bisulfite treatment and the PCR. Bisulfite treatment involves the conversion of unmethylated cytosines to uracils. These uracils are then converted to thymines during PCR thereby making it possible to identify unmethylated sites during DNA sequencing. Bisulfite treatment can potentially degrade DNA, and may result in low quality DNA for sequencing and bias in conversion [20]. However, a new enzymatic-based method, which maintains DNA integrity, has recently been developed [21]. This method works in two parts. First, methylated cytosines are converted into products that cannot be deaminated. Second, unmethylated cytosines are converted to uracils. Both steps are carried out with enzymes and allow single base resolution of DNA methylation similar to bisulfite treatment. As DNA integrity is maintained, smaller quantities of starting DNA are required for this process [22–24]. This makes it desirable in settings where large quantities of DNA from individuals are difficult to collect or DNA is needed for multiple analyses.

In our previous studies, the PCR step for amplifying the age-associated loci has involved the handling of multiple PCR additives [18,19,25]. By handling each individual PCR additive, it is possible to make minor adjustments to the reaction to ensure proper amplification of the desired PCR product. However, manual handling of each reagent makes this assay susceptible to errors which may cause batch effects and are difficult to automate on liquid handling robots. Therefore, a commercial PCR master mix, whereby the only components to handle are the DNA, primers, and the master mix itself would be advantageous to reduce complexity. In this study, we aimed to improve an existing epigenetic ageing assay for the Australian Lungfish (*Neoceratodus forsteri*) [18] as a proof-of-concept for an improved workflow. This existing assay used a bisulfite treatment protocol and a multiplex PCR that contained handling of individual PCR additives. We compared a commercial bisulfite treatment approach with an enzymatic approach for DNA methylation detection and used a commercial PCR master mix to streamline the workflow. By using a known age data set of lungfish, we could detect if any modifications in the protocol resulted in differences in age prediction.

## Methods

### Animal ethics

The Australian Lungfish samples used in this study have been used in previous studies, a total of 96 were used in this study. The samples were collected from the Brisbane, Burnett, and Mary rivers in Queensland, Australia. General Fisheries Permits 174232 and 140615 were approved by Australian Ethics Committee protocol numbers CA2011/10/551 and ENV/17/14/AEC. Fin tissue DNA was extracted using the DNeasy Blood & Tissue Kit (Qiagen, Cat. 69504) following the manufacturer's protocol.

### Bisulfite treatment and enzymatic deamination

Both bisulfite treatment and enzymatic deamination were carried out with commercial kits. Bisulfite treatment was carried using the Zymo EZ-96 DNA Methylation-Gold Kit (Cat. D5008) with a total of 500 ng of genomic DNA. Enzymatic deamination was performed using a total of 200 ng of genomic DNA with the NEBNext® Enzymatic Methyl-seq Conversion Module (Cat. E7125L). Both commercial kits were carried out without any deviation.

## Multiplex PCR and DNA sequencing

Primers previously used [18] were modified to include the Illumina nextera-overhang sequences (S1 Table in S1 File). To simplify the master mix for large scale projects, a commercial PCR master mix (Platinum™ SuperFi™ U Multiplex Master Mix (Invitrogen Cat. A5140096) was used, whereas in previous studies a PCR master mix was made using the Promega 5x Green GoTaq (Cat. M891A) and additional PCR additives [18]. For the multiplex PCR, the manufacturer's master mix (S2 Table in S1 File) and cycling conditions (S3 Table in S1 File) were used. Illumina unique-dual indexing barcodes (Cat. 20025019) were added during the secondary PCR reaction (S4 and S5 Tables in S1 File). PCR products were cleaned after both the multiplex and barcoding reactions with Solid Phase Reverse Immobilization (SPRI) bead mix (see supplementary information for full details). Cleaned PCR products were pooled together in equal volumes after the barcoding reaction. The product was checked using an Aligent Tapestation (Cat. D1000, S1 Fig in S1 File and S1 Raw image). DNA sequencing was carried out at the Australian Genome Research Facility (AGRF) with an Illumina MiSeq with V2 chemistry and 150 paired end reads (300 cycles) with the aim to generate 100,000 reads per sample. Identical protocols for multiplex PCR and DNA sequencing were applied to busulfite and enzymatic treated DNA. The bisulfite and enzymatic treated samples were pooled into separate libraries and sequenced separately as the same set of barcodes were being used.

## Data analysis

Sequencing quality of each sample was obtained using fastqc [26]. Sequencing data was hard clipped with seqkit v1.2 at the 5' and 3' end by 15bp to remove adaptor sequences [27]. Reads were aligned to the lungfish reference genome (assembly neoFor_v3) using Bismark v0.20.0 with default parameters and bowtie2 as an aligner [28,29]. Methylation calling was carried out using the bismark_methylation_extractor function and values were returned as a percentage. Epigenetic age was determined using the previously developed model which was calibrated on a larger data set [18]. The epigenetic age was compared to the known age of the individuals using Pearson correlations and absolute error rates. Differences between the bisulfite and enzymatic treatments were compared using Kolmogorov-Smirnov for sequencing quality, and paired t-tests as the same set of samples were used in both methods. Sequencing quality was compared by using average Q-scores across the read generated from fastqc [26,30]. All statistical analyses were carried out in R version 4.2.0 [31].

## Results

### Sequencing quality

Comparison between the two libraries found the enzymatic treated library to have a higher sequencing quality average across the read compared to the bisulfite treated library (Kolmogorov-Smirnov, p-value < 0.01, Fig 1A). Total reads per sample was also found to be significantly higher in the bisulfite (106,207 reads) treated library compared to the enzymatic (89,544 reads) treated library (paired t-test, p-value < 0.01, Fig 1B). As reflected in Fig 1B, the standard deviation of total reads was found to be greater in the bisulfite treated library. This too was found across all amplicons (Fig 2). A total of six samples were excluded in the bisulfite library as they did not have any coverage across multiple amplicons. This made age prediction impossible in these samples. In comparison, the enzymatic method had sufficient coverage across all amplicons (minimum coverage: 500 reads).

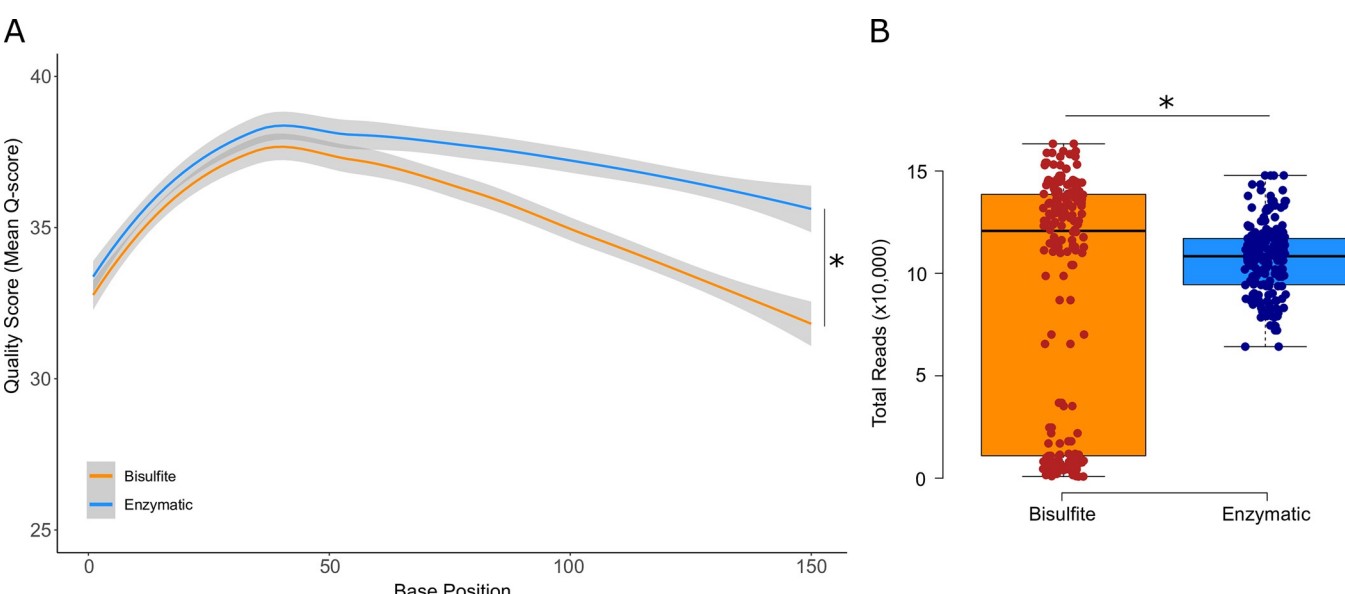

**Fig 1.** Sequencing quality between the bisulfite and enzymatic treated DNA over the **A.** length of the read (* Kolmogorov-Smirnov, p-value < 0.01) and **B.** total reads produced per sample in each library (* paired t-test, p-value < 0.01).

## Epigenetic age prediction

Epigenetic age prediction in both the bisulfite (Pearson correlation = 0.99, p-value < 0.01) and enzymatic (Pearson correlation = 0.99, p-value < 0.01) treated DNA libraries were found to have a high correlation between the known and predicted age (Fig 3A and 3B). The absolute

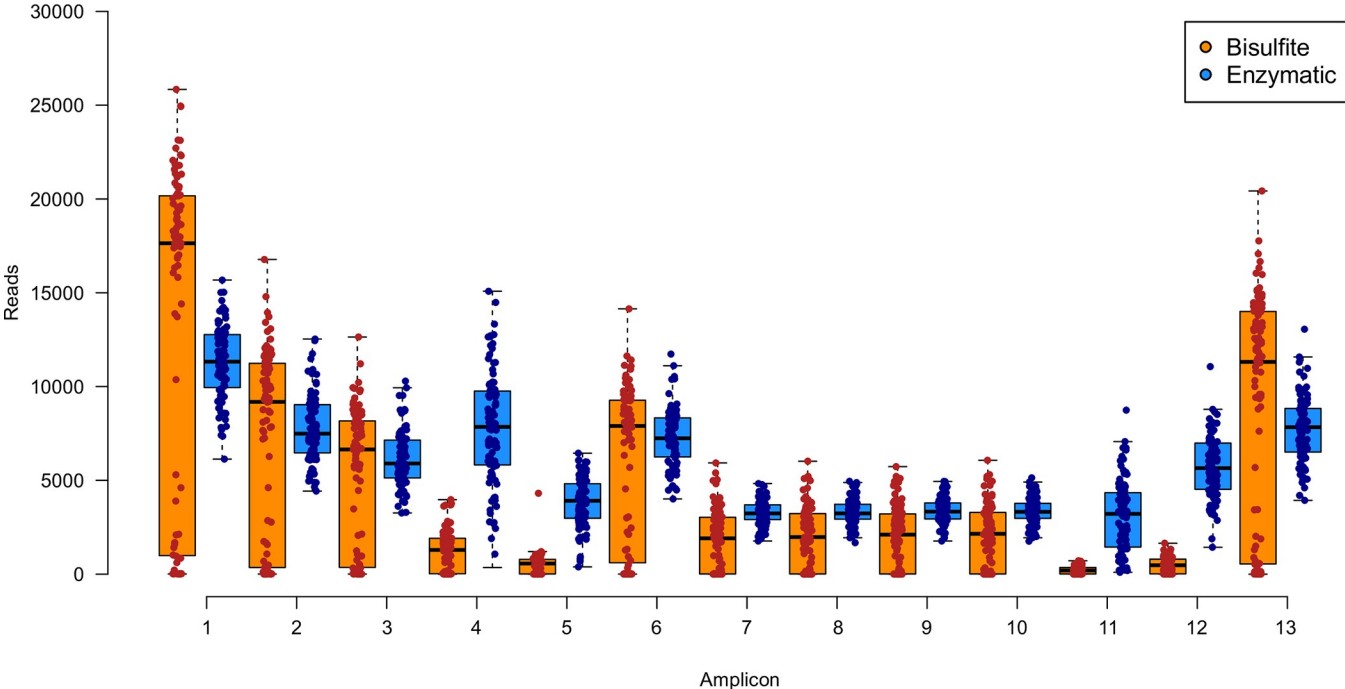

**Fig 2. Reads per amplicon between each the bisulfite and enzymatic treated DNA.** Each dot represents an individual sample and are coloured whether it was bisulfite or enzymatic treated. The dark horizontal line in each box whisker plot represents the median.

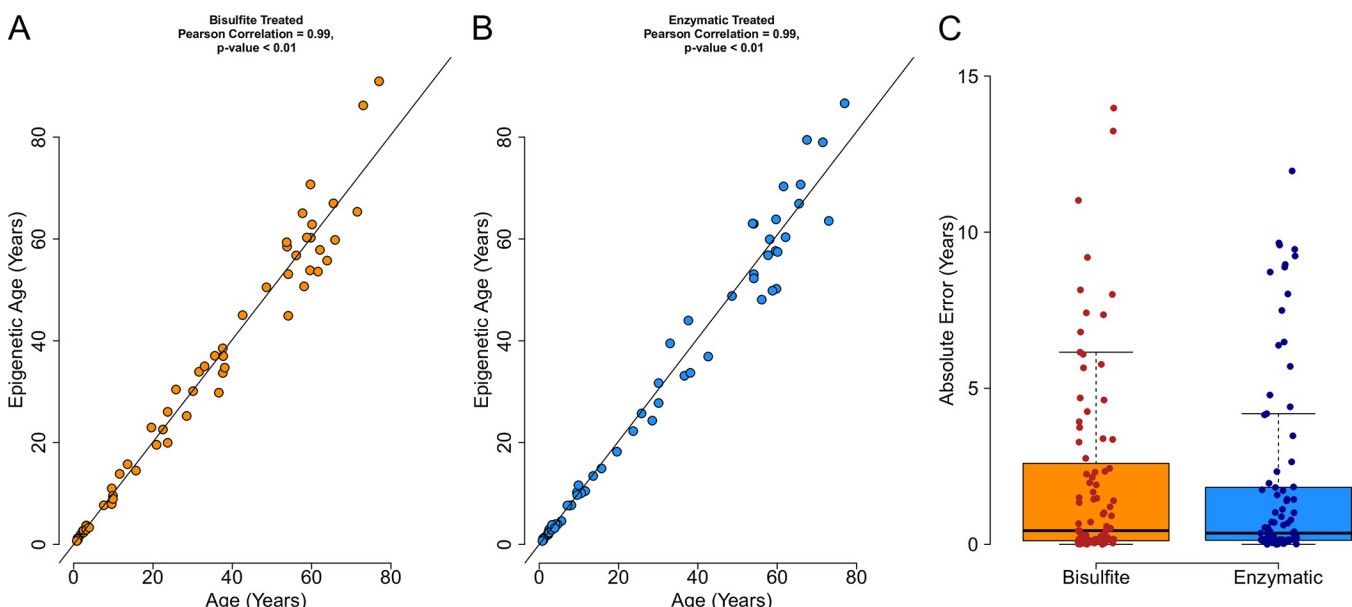

**Fig 3.** Epigenetic age prediction of the Australian Lungfish with either **A.** bisulfite or **B.** enzymatic treated DNA with a commercial PCR master mix used for both. **C.** Absolute error rates between each library. No statistical difference was found between each data set (paired t-test, p-value = 0.69).

error rates between each library were found to be not significantly different (paired t-test, p-value = 0.69, Fig 3C). Compared to the original assay which used an in-house bisulfite protocol [18], no statistical difference was observed in both the bisulfite protocol with the commercial kit in this study and enzymatic methods in absolute error (unpaired t-tests, p-values > 0.05). Both the bisulfite and enzymatic age predictions have a high correlation (Pearson correlation = 0.98, p-value < 0.01, S2 Fig in S1 File) suggesting both methods produce reliable age predictions.

### Reagent handling

The bisulfite (total of eight reagents) and enzymatic treatments (total of ten reagents) did not differ greatly between the total number reagents (S6 Table in S1 File). However, the comparison between the in-house protocol for the multiplex PCR master mix [18] was 8 reagents fewer compared to the commercial master mix used in this study (S6 Table in S1 File).

At the time of writing the manuscript (October 2023), the Zymo EZ-96 DNA Methylation-Gold Kit (Cat. D5008) cost $6.6 AUD per sample and was able to process 96 samples in 3 hours, whereas the NEBNext® Enzymatic Methyl-seq Conversion Module (Cat. E7125L) cost $13.4 AUD per sample and was able to process 96 samples in 6 hours. These cost and time comparisons may change in the future as technology advancements occur and availability of reagents.

### Discussion

Epigenetic age prediction is valuable in biomedical diagnostics and research and for wildlife management, but its widespread adoption is dependent upon both the ease of the assay, and the accuracy of the model to predict the age of the animal. The main limitation with developing an epigenetic clock is the calibration data set. Epigenetic clocks need to be calibrated with known age animals. Unfortunately, the species that need a molecular based method for age prediction, most likely do not have any known age information. Therefore, most epigenetic

clocks most likely do not have an adequate sample size for calibration. However, there are still other aspects of epigenetic clocks such as the assay itself that need further improvement to be widely adopted. In this study, we have improved the sequencing quality for an established epigenetic ageing clock by using an enzymatic method and reduced the number of reagents required by using a commercial PCR master mix. These alternative and simplified approaches improve the overall assay, thus making it more feasible for large scale applications.

The advantage of the enzymatic method over bisulfite sequencing is that it produces less DNA degradation [32], and this is clearly demonstrated by our results (Fig 1). DNA degradation can be a significant issue in the collection of genetic samples from wildlife [33]. Therefore, an already degraded DNA sample may be unable to be sequenced after bisulfite treatment as the cumulation of DNA degradation from both the original sample and the bisulfite treatment may be too high. This makes the enzymatic method more practical when working with wildlife genetic samples that are at risk of DNA degradation because they are inevitably collected in the field, and often by non-experts, and sometimes under sub-optimal and challenging conditions. The enzymatic method also can work with less starting DNA, down to quantities as low as picograms of DNA [21]. This is another advantage for wildlife research, which often operates under tight animal welfare conditions, and in remote settings, where obtaining a high quantity of DNA can be difficult. By requiring less DNA and without degradation during the process, the enzymatic method is well-suited to characterising DNA methylation in genetic wildlife samples. In contrast to the bisulfite treatment, where in our experiment six samples were excluded from the analysis due to no coverage of amplicons, there was no drop out of samples with the enzymatic method with accurate age prediction. The drop out we observed could have resulted from several factors, but DNA degradation during bisulfite treatment is a strong candidate. Drop out of coverage could be overcome by splitting the amplicons into two separate reactions. However, this would require more DNA and would be counter to the desire for simpler laboratory workflows. Other possibilities would most likely be method specific with the bisulfite protocol. PCR showed amplified products on a gel, but because of a multiplex PCR, it is impossible to determine if all amplicons amplified. Potentially there could be further optimisation required to amplify all amplicons, such as modifying the ratios of primers in the PCR master mix. Our results indicate that an enzymatic method offers significant advantages over traditional bisulfite sequencing where certainty of obtaining results is important or where sample quality and quantity may be variable, as may be the case for wildlife samples.

As epigenetic ageing assays become more readily used in wildlife management there is a greater need to increase throughput, especially within fisheries where in the order of 1000s of samples could be processed. It is ideal to reduce the complexity of these assays to reduce batch effects and human error. In this study, we show a commercial PCR master mix performed as well as an existing customized master mix on both bisulfite and enzymatic treated DNA. The disadvantage to using a commercial PCR kit is that the make-up of the contents may be unknown. This can make it difficult in identifying the ideal candidate master mix for the PCR. Therefore, multiple commercial master mixes may need to be tested, thereby making the initial optimisation time-consuming. However, the benefit is that the ongoing assay is more simplified than a customised one.

Epigenetic age prediction is being used more in a variety of research applications. However, for it to be adopted more widely, especially for non-model organisms, the methods need to be more robust and simplified. In this study, we have improved a previous epigenetic age prediction assay by increasing the sequencing quality using an enzymatic method for DNA methylation and simplified the PCR by using a commercial master mix. Cost per sample remains a challenge outside of the biomedical field and currently the enzymatic is approximately double the cost. However, this may become comparable to bisulfite treatment with time. Depending

on a researcher's experiment and budget will determine if a bisulfite or enzymatic approach will be used. The addition of alternatives epigenetic research such as the enzymatic based method for DNA methylation may enable novel research. However, future research should focus on increasing assay efficiency. For example, making DNA sequencing higher throughput to process a larger number of samples or designing additional of i5/i7 indexes. With more improvement in the assays, we anticipate that epigenetic age prediction will become a routine methodology for a wide variety of applications.

## Supporting information

**S1 File. Supplementary tables and figures.**
(DOCX)

**S1 Raw image.**
(PNG)

## Acknowledgments

The authors would like to thank the Australian Research Council project (LP130100118) team including Mark Kennard, Stewart Fallon, Sharon Marshall, Andrew McDougall, Peter Kind, Jane Hughes, Dan Schmidt, and Nick Bond for approving provision of samples. AGRF is supported by the Australian Government National Collaborative Research Infrastructure Initiative through Bioplatforms Australia. We would also like to thank Yi Jin Liew for internal review of our manuscript.

## Author Contributions

**Conceptualization:** Benjamin Mayne, David Chandler, Christopher Noune.

**Data curation:** Thomas Espinoza, David Roberts.

**Formal analysis:** Benjamin Mayne.

**Methodology:** Benjamin Mayne, David Chandler, Christopher Noune, David Roberts, Chloe Anderson.

**Software:** Benjamin Mayne.

**Supervision:** Oliver Berry.

**Writing – original draft:** Benjamin Mayne.

**Writing – review & editing:** David Chandler, Christopher Noune, Thomas Espinoza, David Roberts, Chloe Anderson, Oliver Berry.

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
