## [Decision Letter · Decision Letter 0]

24 Sep 2023

PONE-D-23-26034Increased scalability and sequencing quality of an epigenetic age prediction assayPLOS ONE

Dear Dr. Mayne,

Thank you for submitting your manuscript to PLOS ONE. After careful consideration, we feel that it has merit but does not fully meet PLOS ONE’s publication criteria as it currently stands. Therefore, we invite you to submit a revised version of the manuscript that addresses the points raised during the review process.

The authors provided and tested an alternative assay for epigenetic ageing particularly for lungfish. My suggestions are as below.

1. PCR master mix is a common practice in molecular biology. It should not be emphasized too much in the Abstract for finding a commercial product suitable for the specific purpose.

2. The Results section is relatively short without enough details. For example, I did not find sample numbers of different experimental groups. Please revise it by adding adequate descriptions.

3. Please make point-by-point responses to the comments of the reviewers.

We look forward to receiving your revised manuscript.

Kind regards,

Xianmin Zhu

Academic Editor

PLOS ONE

Journal Requirements:

 "This study was funded by the CSIRO"

Reviewers' comments:

Reviewer's Responses to Questions

**Comments to the Author**

1. Is the manuscript technically sound, and do the data support the conclusions?

Reviewer #1: Yes

Reviewer #2: Yes

Reviewer #3: Yes

2. Has the statistical analysis been performed appropriately and rigorously? 

Reviewer #1: Yes

Reviewer #2: Yes

Reviewer #3: Yes

3. Have the authors made all data underlying the findings in their manuscript fully available?

Reviewer #1: Yes

Reviewer #2: Yes

Reviewer #3: Yes

4. Is the manuscript presented in an intelligible fashion and written in standard English?

Reviewer #1: Yes

Reviewer #2: Yes

Reviewer #3: Yes

5. Review Comments to the Author

Reviewer #1: The manuscript by Mayne et al. 2023 “Increased scalability and sequencing quality of an epigenetic age prediction assay” compares the effect of two different library preparations on the epigenetic age predictions for the Australian lungfish. The project was well executed and represents an important step in integrating epigenetic clocks in ecological research. I think this manuscript can be improved by including estimates for price and time required for each of these library preparations. I have also outlined specific comments below which will add clarity to the manuscript:

Specific comments:

Line 23: Given the applications in conservation ecology, consider adding the price/time difference between this method and previous methods.

Line 27-28: change text to “age related diseases”

Line 35: may be worth citing research demonstrating the utility of using epigenetic clocks designed for one species (i.e. humans) on species with variable evolutionary divergence (i.e. chimpanzees, gorillas[1])

1. Horvath S. DNA methylation age of human tissues and cell types. Genome Biol. 2013;14: 115. Available: http://genomebiology.com//14/10/R115

Line 70: add number of lungfish samples used.

Methods: Instead of grouping methods by the “Bisulfite treatment and Enzymatic deamination” and “Multiplex PCR and DNA sequencing”, I would recommend fully describing the previously established method and the new method in their own sections. It would make it easier to distinguish which text applies to which method.

Line 106 & 110-111: what metric did you use to assess sequencing quality?

Line 107: add citation for R.

Line 114: add “,” after text “As reflected in Figure 1B”

Line 122-123: do these statistics refer to one library type or both? If both, it would be useful to report for the individual library types so the reader can compare performance of each method

Line 125: what is meant by “bisulfite in this study”?

Line 129: what is the cost/time difference between the two methods?

Line 149: replace “inevitable” with ‘inevitably”

Line 157-158: what are the other possible reasons for this? Are they also method specific?

Line 179-180: comparing cost per sample in the methods described here would be useful

Discussion: it would be worth addressing the barriers involved with creating the initial epigenetic clock for a given wildlife species (after which these targeted methods can be applied)

Figures:

Figure 1: In lines 114-115 you state that the enzymatic libraries have greater reads per sample according to their means (t-test). However, figure 1B appears to show higher median reads per sample of the bisulfite libraries (assuming the line in your box plots is the median.) I would clarify which metric you compare statistically and show that metric on the plot. Please also add these details to the figure legend. Also, are the samples which did not pass the threshold for coverage for multiple amplicons included the statistical analysis? I would be interested to know if the mean of reads per sample still differs when excluding the failed samples.

Figure 2: Is there a reason some amplicons (i.e. amplicons 5,11,12) have such low coverage in the bisulfite treatment when compared to the others? Also please add details about what is represented by box plot boxes and lines.

Reviewer #2: The main claim of the paper is that a new version of epigenetic clock used to predict the age in three fish species is superior to previous versions due to the use of more accurate enzyme-based cytosine methylation sequencing method instead of bisulfite sequencing and of a commercial kit optimized for multiplex PCR instead of home-optimized PCR mixtures. These advantages that seem not to be very significant in laboratory setup may become highly beneficial in wild-life context when both the quality and quantity of DNA samples could be rather low. I view this claim as essentially fair except for few small points.

(i) The use of commercial PCR kits is a general practice. Therefore, it hardly deserves a special note in the abstract.

(ii) The statement “Unfortunately, bisulfite treatment degrades DNA, which can result in low quality DNA for sequencing and bias in conversion” (Introduction lines 45-46) looks somewhat exaggerated these days. DNA degradation was a serious problem in the early years of bisulfite conversion technology (referenced by a 2007 year paper [20]) but it is mostly overcome in modern BS conversion kits, such as EpiTect Bisulfite Kit from Qiagen or EZ DNA Methylation-Gold™ Kit from Zymo Research.

(iii) On lines 55-56 of Introduction, authors state that “In previous studies, the PCR step for amplifying the age-associated loci has involved the handling of multiple PCR additives [18, 19, 25]”. It would be more correctly to write “In our previous studies…” since quite a lot of other people (including myself) use commercial PCR kits, such as NEBNext Q5U Master Mix (included in the NEB NEBNext® Enzymatic Methyl-seq Kit).

(iv) Full name of the BS conversion kit used should be indicated on line 77 – EZ-96 DNA Methylation-Gold Kit (Zymo Research Cat. D5008). Otherwise, the Methods section is fine. The methods used are adequate and described in enough detail.

(v) In the authors’ hands, EM-seq libraries show better sequencing results compared with BS-seq libraries. As a matter of fact, similar finding was earlier described in a much more detailed and thorough study by Steve Jacobsen lab (reference 22). However, both EM-seq and BS-seq produce results of fair quality. This is especially true for CpG-specific methylation which is the only concern in most studies. Likewise, the read quality in the BS-seq is quite good, though not as good as in EM-seq (Fig.1A), while the age-prediction looks equally good in EM-seq and BS-seq based assays (Fig.3A,B). As concerning the comparison of job intensities between EM-seq and BS-seq protocols (lines 130-133), it should be noted that in fact the BS-seq protocol involves significantly less hands-on time, and it is more streamlined and robust, and probably significantly cheaper, when the commercial BS conversion kits from Qiagen or Zymo Research are used. Taking into account all these aspects, I would say that the choice between EM-seq and BS-seq in most cases is a matter of personal preference.

The manuscript is well written and easy to understand. As a matter of fact I have no major concerns both about its content and quality. Few points noted above reflect my personal experience and views and as such are somewhat subjective. I do not demand the authors to agree with them, but it would be fair to discuss these points in more detail. With these minor corrections the manuscript could be accepted for publication in PLOS ONE.

Reviewer #3: The manuscript describes a methods to improve the DNA methylation analyis in epigenetic ageing studies.

Authors proposed a simple method where bisulfite treatment and a new enzymatic-based method were compared.

The manuscript is well written, the introduction provides a good background of the subject matter and experimental apparatus is appropriate for the study. Materials and methods are well described, even if the number of samples analyzed or the number of libraries sequenced are not reported, therefore is difficult to evaluate the robustness of the results. I suggest to indicate this information in the text. Moreover, at rows 132-133 Authors states that “the comparison between the in-house protocol for the multiplex PCR master mix [18] was 7 reagents fewer compared to the commercial master mix used in this study (Supplementary Table 6).” But, as reported in Supplementary Table 6, the number of reagents of the in-house protocol for the multiplex PCR master mix was higher than those of the commercial master mix…11 vs 3.

6. PLOS authors have the option to publish the peer review history of their article (what does this mean?). If published, this will include your full peer review and any attached files.

Reviewer #1: No

Reviewer #2: **Yes: **Vasily Ashapkin

Reviewer #3: No

---

## [Author Response · Author response to Decision Letter 0]

19 Nov 2023

Dear Dr. Mayne,

Thank you for submitting your manuscript to PLOS ONE. After careful consideration, we feel that it has merit but does not fully meet PLOS ONE’s publication criteria as it currently stands. Therefore, we invite you to submit a revised version of the manuscript that addresses the points raised during the review process.

Response: We would like to thank both editors and reviewers for their time and consideration of our manuscript. Below we have provided our response to the comments and suggestions and we look forward to hearing a response from the journal.

Thank you,

Benjamin Mayne

The authors provided and tested an alternative assay for epigenetic ageing particularly for lungfish. My suggestions are as below.

1. PCR master mix is a common practice in molecular biology. It should not be emphasized too much in the Abstract for finding a commercial product suitable for the specific purpose.

Response: We’ve removed the mention of the PCR mastermix as per the suggestion and since the manuscript is more focused on the use of the enzymatic approach. 

2. The Results section is relatively short without enough details. For example, I did not find sample numbers of different experimental groups. Please revise it by adding adequate descriptions.

Response: We’ve added the sample numbers and have also added a more detailed description of the original assay as per reviewer 1’s suggestions. 

3. Please make point-by-point responses to the comments of the reviewers.

Response: A point-by-point response is provided below. 

Reviewer #1: The manuscript by Mayne et al. 2023 “Increased scalability and sequencing quality of an epigenetic age prediction assay” compares the effect of two different library preparations on the epigenetic age predictions for the Australian lungfish. The project was well executed and represents an important step in integrating epigenetic clocks in ecological research. I think this manuscript can be improved by including estimates for price and time required for each of these library preparations. I have also outlined specific comments below which will add clarity to the manuscript:

Response: Thanks for the time to review our manuscript. We have provided cost comparisons into the manuscript and have expanded the methods to detail the previous method. 

Specific comments:

Line 23: Given the applications in conservation ecology, consider adding the price/time difference between this method and previous methods.

Response: We have provided a cost comparison by just saying it is approximately doubled, but have provided more detailed comparisons in the Results in the Reagent Handling section. 

Line 27-28: change text to “age related diseases”

Response: Changed text. 

Line 35: may be worth citing research demonstrating the utility of using epigenetic clocks designed for one species (i.e. humans) on species with variable evolutionary divergence (i.e. chimpanzees, gorillas[1]) 1. Horvath S. DNA methylation age of human tissues and cell types. Genome Biol. 2013;14: 115. Available: http://genomebiology.com//14/10/R115

Response: We’ve added the citation and a sentence discussing the issue of implementing a similar array for other species on line 35. 

Line 70: add number of lungfish samples used.

Response: Added the number of samples used on the same line. 

Methods: Instead of grouping methods by the “Bisulfite treatment and Enzymatic deamination” and “Multiplex PCR and DNA sequencing”, I would recommend fully describing the previously established method and the new method in their own sections. It would make it easier to distinguish which text applies to which method.

Response: We are unsure what aspects were ambiguous to the reviewer. We have left the section Bisufite Teamtment and Enzymatic Deamination unchanged except for correcting the name of a kit. We have clarified in the Multiplex PCR and DNA Sequencing that those processes were identical for DNA processed via bisulfite or enzymatic workflows (line 101).

Line 106 & 110-111: what metric did you use to assess sequencing quality?

Response:We used average q-scores across the read. This has been added to the methods in the Data Analysis section. 

Line 107: add citation for R.

Response: Citation added

Line 114: add “,” after text “As reflected in Figure 1B”

Response: Comma added

Line 122-123: do these statistics refer to one library type or both? If both, it would be useful to report for the individual library types so the reader can compare performance of each method

Response: It was both, this has been made more clear in the manuscript. 

Line 125: what is meant by “bisulfite in this study”?

Response: We were referring to the bisulfite protocol with the commercial kit that was carried out in the study. The sentence now reads ‘Compared to the original assay which used an in-house bisulfite protocol [18], no statistical difference was observed in both the bisulfite protocol with the commercial kit in this study…’

Line 129: what is the cost/time difference between the two methods?

Response: Here we have added the current cost/samples and time to process 96 samples. The money values are within AUD currency. 

Line 149: replace “inevitable” with ‘inevitably”

Response: Fixed.

Line 157-158: what are the other possible reasons for this? Are they also method specific?

Response: Potentially there could be further optimisation needed with the PCR protocol such as different ratios of the primers to get the protocol to amplify all amplicons with the bisuflite protocol. THis has been added into lines 161-164.

Line 179-180: comparing cost per sample in the methods described here would be useful

Response: We inserted a line stating the cost comparison between the bisulfite and enzymatic approach. 

Discussion: it would be worth addressing the barriers involved with creating the initial epigenetic clock for a given wildlife species (after which these targeted methods can be applied)

Response: The main barrier was obtaining a calibration data set. We’ve now added this into the first paragraph of the discussion. 

Figures:

Figure 1: In lines 114-115 you state that the enzymatic libraries have greater reads per sample according to their means (t-test). However, figure 1B appears to show higher median reads per sample of the bisulfite libraries (assuming the line in your box plots is the median.) I would clarify which metric you compare statistically and show that metric on the plot. Please also add these details to the figure legend. Also, are the samples which did not pass the threshold for coverage for multiple amplicons included the statistical analysis? I would be interested to know if the mean of reads per sample still differs when excluding the failed samples.

Response: Apologies, that was an error on lines 114-115 it should be the other way round and has been corrected. 

Figure 2: Is there a reason some amplicons (i.e. amplicons 5,11,12) have such low coverage in the bisulfite treatment when compared to the others? Also please add details about what is represented by box plot boxes and lines.

Response: We’re unsure why some amplicons had lower coverage than others. Potentially further optimisation is required and as now discussed in the discussion on lines 177-181 this could be done my modifying the primer ratios in the master mix. Details of the box whisker plot have been added into the figure legend.

Reviewer #2: The main claim of the paper is that a new version of epigenetic clock used to predict the age in three fish species is superior to previous versions due to the use of more accurate enzyme-based cytosine methylation sequencing method instead of bisulfite sequencing and of a commercial kit optimized for multiplex PCR instead of home-optimized PCR mixtures. These advantages that seem not to be very significant in laboratory setup may become highly beneficial in wild-life context when both the quality and quantity of DNA samples could be rather low. I view this claim as essentially fair except for few small points.

Response: Thank you for your time to review our manuscript. We have provided details to your suggestions below.

(i) The use of commercial PCR kits is a general practice. Therefore, it hardly deserves a special note in the abstract.

Response: Mention of the use of a commercial PCR kit have been removed from the abstract. 

(ii) The statement “Unfortunately, bisulfite treatment degrades DNA, which can result in low quality DNA for sequencing and bias in conversion” (Introduction lines 45-46) looks somewhat exaggerated these days. DNA degradation was a serious problem in the early years of bisulfite conversion technology (referenced by a 2007 year paper [20]) but it is mostly overcome in modern BS conversion kits, such as EpiTect Bisulfite Kit from Qiagen or EZ DNA Methylation-Gold™ Kit from Zymo Research.

Response: The sentence has been modified to as below to reduce any potential exaggeration in DNA degradation – noting, however, that the results in our manuscript do clearly identify that busulfite converted DNA template was degraded relative to enzymatically converted DNA template.

‘Bisulfite treatment can potentially degrade DNA, and may result in low quality DNA for sequencing and bias in conversion’ 

(iii) On lines 55-56 of Introduction, authors state that “In previous studies, the PCR step for amplifying the age-associated loci has involved the handling of multiple PCR additives [18, 19, 25]”. It would be more correctly to write “In our previous studies…” since quite a lot of other people (including myself) use commercial PCR kits, such as NEBNext Q5U Master Mix (included in the NEB NEBNext® Enzymatic Methyl-seq Kit).

Response: We’ve changed the start of the sentence to ‘In our previous studies’

(iv) Full name of the BS conversion kit used should be indicated on line 77 – EZ-96 DNA Methylation-Gold Kit (Zymo Research Cat. D5008). Otherwise, the Methods section is fine. The methods used are adequate and described in enough detail.

Response: Thank you and we have updated the name of the kit. 

(v) In the authors’ hands, EM-seq libraries show better sequencing results compared with BS-seq libraries. As a matter of fact, similar finding was earlier described in a much more detailed and thorough study by Steve Jacobsen lab (reference 22). However, both EM-seq and BS-seq produce results of fair quality. This is especially true for CpG-specific methylation which is the only concern in most studies. Likewise, the read quality in the BS-seq is quite good, though not as good as in EM-seq (Fig.1A), while the age-prediction looks equally good in EM-seq and BS-seq based assays (Fig.3A,B). As concerning the comparison of job intensities between EM-seq and BS-seq protocols (lines 130-133), it should be noted that in fact the BS-seq protocol involves significantly less hands-on time, and it is more streamlined and robust, and probably significantly cheaper, when the commercial BS conversion kits from Qiagen or Zymo Research are used. Taking into account all these aspects, I would say that the choice between EM-seq and BS-seq in most cases is a matter of personal preference.

Response: We ultimately agree with the reviewer that it may come down to personal choice, most likely a combination of the experiment itself and budget. We have added these sentences into the concluding paragraph discussing this.

‘Depending on a researcher’s experiment and budget will determine if a bisulfite or enzymatic approach will be used. The addition of alternatives epigenetic research such as the enzymatic based method for DNA methylation may enable novel research.’

The manuscript is well written and easy to understand. As a matter of fact I have no major concerns both about its content and quality. Few points noted above reflect my personal experience and views and as such are somewhat subjective. I do not demand the authors to agree with them, but it would be fair to discuss these points in more detail. With these minor corrections the manuscript could be accepted for publication in PLOS ONE.

Response: We have made all changes as suggested by the reviewer and thank them for their suggestions in improving the overall manuscript. 

Reviewer #3: The manuscript describes a methods to improve the DNA methylation analyis in epigenetic ageing studies.

Authors proposed a simple method where bisulfite treatment and a new enzymatic-based method were compared.

The manuscript is well written, the introduction provides a good background of the subject matter and experimental apparatus is appropriate for the study. Materials and methods are well described, even if the number of samples analyzed or the number of libraries sequenced are not reported, therefore is difficult to evaluate the robustness of the results. I suggest to indicate this information in the text. Moreover, at rows 132-133 Authors states that “the comparison between the in-house protocol for the multiplex PCR master mix [18] was 7 reagents fewer compared to the commercial master mix used in this study (Supplementary Table 6).” But, as reported in Supplementary Table 6, the number of reagents of the in-house protocol for the multiplex PCR master mix was higher than those of the commercial master mix…11 vs 3.

Response: We would like to thank the reviewer for their time on our manuscript.

We added the sample numbers and libraries to the manuscript and have fixed the comparison between the in-house and commercial master mix reagents.

---

## [Decision Letter · Decision Letter 1]

27 Dec 2023

Increased scalability and sequencing quality of an epigenetic age prediction assay

PONE-D-23-26034R1

Dear Dr. Mayne,

We’re pleased to inform you that your manuscript has been judged scientifically suitable for publication and will be formally accepted for publication once it meets all outstanding technical requirements.

Kind regards,

Xianmin Zhu

Academic Editor

PLOS ONE

Additional Editor Comments (optional):

Reviewers' comments:

Reviewer's Responses to Questions

**Comments to the Author**

1. If the authors have adequately addressed your comments raised in a previous round of review and you feel that this manuscript is now acceptable for publication, you may indicate that here to bypass the “Comments to the Author” section, enter your conflict of interest statement in the “Confidential to Editor” section, and submit your "Accept" recommendation.

Reviewer #1: All comments have been addressed

Reviewer #2: (No Response)

Reviewer #3: All comments have been addressed

2. Is the manuscript technically sound, and do the data support the conclusions?

Reviewer #1: Yes

Reviewer #2: Yes

Reviewer #3: (No Response)

3. Has the statistical analysis been performed appropriately and rigorously? 

Reviewer #1: Yes

Reviewer #2: Yes

Reviewer #3: (No Response)

4. Have the authors made all data underlying the findings in their manuscript fully available?

Reviewer #1: Yes

Reviewer #2: Yes

Reviewer #3: (No Response)

5. Is the manuscript presented in an intelligible fashion and written in standard English?

Reviewer #1: Yes

Reviewer #2: Yes

Reviewer #3: (No Response)

6. Review Comments to the Author

Reviewer #1: The authors have addressed all of the points brought up in my initial review. I think this manuscript will be a nice contribution to the growing interest in epigenetic clocks for ecological research and conservation.

Reviewer #2: (No Response)

Reviewer #3: (No Response)

7. PLOS authors have the option to publish the peer review history of their article (what does this mean?). If published, this will include your full peer review and any attached files.

Reviewer #1: No

Reviewer #2: **Yes: **Vasily Ashapkin

Reviewer #3: No

---

## [Editor Report · Acceptance letter]

2 May 2024

PONE-D-23-26034R1 

PLOS ONE

Dear Dr. Mayne, 

I'm pleased to inform you that your manuscript has been deemed suitable for publication in PLOS ONE. Congratulations! Your manuscript is now being handed over to our production team.

Kind regards, 

on behalf of

Dr. Xianmin Zhu 

Academic Editor

PLOS ONE